# *N*-Acetylcysteine (NAC): Impacts on Human Health

**DOI:** 10.3390/antiox10060967

**Published:** 2021-06-16

**Authors:** Micaely Cristina dos Santos Tenório, Nayara Gomes Graciliano, Fabiana Andréa Moura, Alane Cabral Menezes de Oliveira, Marília Oliveira Fonseca Goulart

**Affiliations:** 1Institute of Chemistry and Biotechnology, Federal University of Alagoas, Maceió 57072-970, Alagoas, Brazil; micaely.tenorio@hotmail.com; 2Institute of Biological and Health Sciences, Federal University of Alagoas, Maceió 57072-970, Alagoas, Brazil; nayaragraciliano@hotmail.com (N.G.G.); alanecabral@gmail.com (A.C.M.d.O.); 3College of Nutrition, Federal University of Alagoas, Maceió 57072-970, Alagoas, Brazil; fabiana.moura@fanut.ufal.br; 4College of Medicine, Federal University of Alagoas, Maceió 57072-970, Alagoas, Brazil

**Keywords:** *N*-acetylcysteine, mechanism of action, antioxidant, anti-inflammatory

## Abstract

*N*-acetylcysteine (NAC) is a medicine widely used to treat paracetamol overdose and as a mucolytic compound. It has a well-established safety profile, and its toxicity is uncommon and dependent on the route of administration and high dosages. Its remarkable antioxidant and anti-inflammatory capacity is the biochemical basis used to treat several diseases related to oxidative stress and inflammation. The primary role of NAC as an antioxidant stems from its ability to increase the intracellular concentration of glutathione (GSH), which is the most crucial biothiol responsible for cellular redox imbalance. As an anti-inflammatory compound, NAC can reduce levels of tumor necrosis factor-alpha (TNF-α) and interleukins (IL-6 and IL-1β) by suppressing the activity of nuclear factor kappa B (NF-κB). Despite NAC’s relevant therapeutic potential, in several experimental studies, its effectiveness in clinical trials, addressing different pathological conditions, is still limited. Thus, the purpose of this chapter is to provide an overview of the medicinal effects and applications of NAC to human health based on current therapeutic evidence.

## 1. Introduction

*N*-Acetylcysteine (NAC) is a drug approved by the Food and Drug Administration (FDA) and recognized by the World Health Organization (WHO) as an essential drug, widely used for the treatment of acetaminophen overdose (paracetamol) and more recently as a mucolytic agent, in respiratory diseases [1]. In some countries, including the United States, Canada, and Australia, NAC is commonly available as an over-the-counter nutritional supplement, with antioxidant properties and great commercial appeal as a nutraceutical [2].

The primary role of NAC is associated with its antioxidant and anti-inflammatory activity, which favors the maintenance of a cellular redox imbalance. For this reason, its therapeutic potential concerns a series of diseases that link oxidative stress to its etiology and progression [3,4]. However, the mechanisms via which NAC exerts its antioxidant and cytoprotective capacity in different physiological conditions have not yet been fully clarified [5]. The growing interest in investigating the favorable effects of NAC involves not only its action as a potent cell bio-protector but also its pharmacokinetic characteristics, related to safety, absorption, and bioavailability, associated with its low cost [3,4].

Animal studies have shown that NAC exerts a potent protective effect against oxidative stress and inflammation under different conditions, including improvement of brain damage induced by transient cerebral ischemia [6], pain and inflammation management in case of infection [7], and restoration of thyroid morphology via reduced infiltration of inflammatory cells [8]. Although several in vivo and ex vivo studies have shown that NAC plays important biological actions that can potentially support its putative therapeutic roles, its effectiveness in clinical studies in addressing different pathological conditions still has conflicting results [4,9]. Thus, the purpose of this review is to provide an overview of the medical effects and applications of NAC to human health based on current therapeutic evidence.

## 2. Pharmacokinetics and Bioavailability

NAC can be administered orally, intravenously, or by inhalation, being commonly safe and well tolerated, even in high doses [10]. Orally, it suffers rapid intestinal absorption and metabolism by the liver, which directs most of the cysteine released toward GSH synthesis [11]. After oral administration, its maximum plasma concentration (*C*_max_) occurs approximately between 1 and 2 h [12].

The bioavailability of free NAC is very low (<10%), and only a tiny amount of the intact molecule reaches the plasma and tissues [13,14]. Additionally, due to the variety of ways that NAC can be found in plasma (oxidized, reduced, and bound to proteins), its pharmacokinetics (PK) is not yet fully understood [15,16]. NAC can be oxidized to a disulfide, *N*,*N*′-diacetylcystine, and it still generates mixed disulfides via a reaction with other low-molecular-weight thiols. After their complete metabolism, cysteine, cystine, inorganic sulfate, and glutathione are the primary metabolic products produced [14].

Due to the absence of the first-pass intestinal and hepatic metabolism, intravenous administration allows rapid delivery of high concentrations of NAC, being the route used for the treatment of paracetamol overdose [16]. After administering intravenous NAC at a dosage of 150 mg/kg over 15 min, the *C*_max_ NAC was on average 554 mg/L [17]. The volume of distribution (Vd) of the total NAC ranges from 0.33 to 0.47 L/kg [13,14].

According to the study by Olsson et al. (1988) [14], after the administration of intravenous NAC (200 mg diluted 1:10 in 0.9% saline), covalent binding to proteins was significant after 60 min. It increased over time to reach a maximum of 50%, 4 h after the dose, decreasing again to approximately 20% after 12 h. For total NAC, the terminal half-life was 5.58 h after intravenous administration and 6.25 h after oral administration of 400 mg. Oral bioavailability was 4.0% and 9.1% for reduced and total NAC, respectively.

In patients with severe liver injury, PK appears to be altered due to less clearance. In the study by Jones et al. (1997) [15], the PK of NAC was evaluated in patients with chronic liver disease versus control. After administering a 600 mg dose of intravenous NAC over 3 min, the area under the serum concentration versus time curve was almost double for cirrhotic patients (152.3 vs. 93.9 mg/L/h) compared to healthy controls. Similarly, patients with end-stage renal disease (ESRD) and normal liver function also have lower total NAC clearance when compared to healthy individuals [18].

The study by Nolin et al. (2010) [18] evaluated the PK of NAC after oral administration of multiple doses (600 mg and 1200 mg of NAC every 12 h) to patients with ERSD versus healthy controls (600 mg every 12 h) for a period of 14 days. Significant dose-related increases in *C*_max_ and plasma concentration–time curve values were observed in patients with ERSD at different dosages, with no change in total clearance. Between patients with ERSD and healthy controls, there were significant PK differences. The full clearance of 600 mg of oral NAC in subjects with ESRD was 4.9 ± 3.5 L/h versus 56.1 ± 12.7 L/h in healthy subjects. Thus, the total clearance of NAC was reduced by 90% in patients with ERSD, with a half-life 13 times greater (51.3 ± 36.7 h versus 3.7 ± 0.8 h control) and, consequently, greater systemic exposure in these individuals.

The concomitant use of NAC orally with activated charcoal can reduce the bioavailability of NAC by impairing its absorption. However, the results are still controversial [16]. The elimination of NAC occurs through the renal system, in which about 30% (27.0 ± 12.8) is excreted in the urine [10] and only 3% is excreted in the feces [19]. Oral administration of 100 mg of ^35^S-labeled NAC to patients with respiratory disorders showed renal excretion of approximately 22% of radioactivity through urine (range 13–38%) after 24 h [20]. A more recent study showed the range of urinary excretion of Chinese (3.66%) and Caucasian (3.80%) subjects after 600 mg of oral NAC administration. All research participants were healthy. According to the authors, the reduced elimination range found in the study was consistent with the extensive metabolism and transformation that NAC undergoes after its administration [21].

The adverse effects of NAC vary from mild to severe and depend on the formulation and dosage used, but an extensive review of multicentric medical records has shown that intravenous and oral NAC is associated with minimal side-effects. In oral administration, the most common adverse effects are gastrointestinal symptoms such as nausea and vomiting, which occur in up to 23% of patients [22]. Other reactions include itching and erythema [11]. The pungent smell of NAC, which resembles rotten eggs (due to sulfur), also contributes to manifestations of nausea and vomiting after oral administration. NAC is commonly diluted in caffeine-free diet sodas to mask the smell and taste and to facilitate acceptance. Effervescent flavored tablets already exist as a new formulation of NAC [23]. Intravenous administration can also cause symptoms such as nausea and vomiting, with a frequency of up to 9% [22].

Considering the formulation, intravenous administration usually causes a more significant proportion of adverse effects when compared to oral administration, especially after the infusion of the initial load, which releases a high plasma concentration of NAC [11]. More serious adverse effects, such as anaphylactoid reactions, are uncommon and more remarkable in up to 8.2% of patients. Anaphylactoid reactions involve a response of nonimmunological origin, probably related to the release of non-IgE-mediated histamine. They include cutaneous symptoms, such as flushing, itching, and angioedema, and systemic symptoms, such as bronchospasm and hypotension. The manifestations of cutaneous symptoms in anaphylactoid reactions are usually greater, with a frequency of 75% [24].

The inhaled form of NAC is most commonly used to treat respiratory diseases. In general, studies with inhaled NAC demonstrated a safety profile similar to other formulations, demonstrating good tolerance to this type of formulation [25,26,27]. Adverse symptoms after administration of inhaled NAC include bacterial pneumonia, cough, sore throat, and drug-induced pneumonitis, among which coughing is the most common [27]. A systematic review with meta-analysis showed that the incidence of adverse effects was significantly higher in the treatment with inhaled NAC when compared to oral NAC; however, the study did not identify any significant difference in the incidence of adverse effects between NAC therapy and the treatments of control [28].

The toxicity of NAC overdose has not yet been defined for patients with paracetamol overdose or for healthy people using single or repeated doses of NAC [29]. It has been reported that an overdose of NAC (100 g) in a short time can cause hemolysis, thrombocytopenia, acute renal failure, and death in patients with glucose-6-phosphate dehydrogenase (G6PD) normal. The reported case resulted from an error during the administration of the loading dose of NAC for the treatment of paracetamol overdose, which should be 10 g (10,000 mg) [30]. Toxicity data for NAC in animals are available [29]. Box 1 provides a summary of the main characteristics of NAC.

Box 1Summary of the compound *N*-acetylcysteine.
**Summary of the Compound *N*-Acetylcysteine**
Drug indicationNAC is used mainly as a mucolytic and in the management of acetaminophen (paracetamol) overdose.Chemical structure
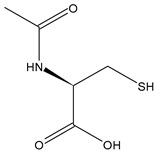
Molecular formulaC_5_H_9_NO_3_SSynonyms
▪*N*-Acetyl-cysteine;▪Acetylcysteine;▪*N*-Acetylcysteine;▪NAC;▪Ac-Cys-OH;▪616–91-1.
pKa3.24: –COOHand 9.5: –SHMolecular weight163.2 g/molProtein binding66–97% (usually to albumin)WHO essential medicinesAntidotes and other substances used in poisonings.Drug classesTherapeutic usesTasteCharacteristic sour taste
▪Antidotes;▪Toxicological emergency.

▪Antiviral agents;▪Expectorants;▪Free-radical scavengers.
Color/FormWhite crystalline powderOdorSlight acetic odorDrug warnings
▪Anaphylactoid reactions, such as rash, hypotension, wheezing, and/or dyspnea, have been reported after administration of parenteral NAC;▪Anaphylactoid reactions can be more severe and even cause death in patients with asthma;▪Skin rash, urticaria, and pruritus are the most reported adverse effects after intravenous NAC administration;▪Acute flushing and erythema are less serious adverse reactions after intravenous administration of NAC;▪Chest tightness and bronchoconstriction have been reported after the use of NAC;▪Bronchospasm clinically induced by NAC is rare;▪Increased airway obstruction can occur after oral NAC inhalation;▪Gastrointestinal symptoms, including nausea and vomiting, are more frequent and may occur after oral administration of NAC.
Absorption
▪Bioavailability is 6–10% following oral administration and less than 3% following topical administration;▪Serum concentrations after intravenous administration of an initial loading dose of 150 mg/kg over 15 min are about 500 mg/L.
Metabolism
▪Deacetylated by the liver to cysteine and subsequently metabolized;▪After oral inhalation or intratracheal instillation, most of the administered drug appears to participate in the sulfhydryl disulfide reaction, while the rest is absorbed by the pulmonary epithelium, deacetylated by the liver to cysteine to be metabolized.
Half-lifeClearanceVolume of distribExcretion
▪Adults—5.6 h;▪Neonates—11 h.
0.11 L/h/kgution0.47 L/kg
▪13–38% (urine);▪3% (feces).
Dosage formsOverdosage
▪Solution for inhalation;▪Intravenous injection;▪Oral solution;▪Effervescent tablets;▪Capsules;▪Tablets;▪Powder;▪Granule;▪Liquid;▪Ocular solution.
Single intravenous doses of NAC that were lethal:
▪1000 mg/kg in mice;▪2445 mg/kg in rats;▪1500 mg/kg in guinea pigs;▪1200 mg/kg in rabbits;▪ 500 mg/kg in dogs.
Symptoms of acute toxicity were ataxia, hypoactivity, labored respiration, cyanosis, loss of righting reflex, and convulsions.Drug–drug interactions
▪NAC can reduce the excretion of various drugs (e.g., pravastatin, valsartan, erythromycin, torasemide, lovastatin, liothyronine, digoxin, cerivastatin, raloxifene, caspofungin, enalapril, simvastatin, penicillamine);▪The serum concentration of eluxadoline can be increased when combined with NAC;▪The therapeutic efficacy of NAC can be decreased when used in combination with oxytetracycline, trypsin, or tetracycline.
Adapted from PubChem (2021) [31] and DrugBank (2021) [32].

## 3. Mechanism of Molecular Action

NAC is a thiol that acts as an acetylated precursor to the amino acid l-cysteine; it can reduce various radicals, by donating one electron, or acts as a nucleophile by donating one or two electrons (Figure 1) [33,34]. Its chemical structure, formed by the sulfhydryl functional group (–SH) plus an acetyl group (–COCH_3_) linked to the amino group (NH_2_), is responsible for its metabolic activities related to the direct and indirect antioxidant action and mucolytic action [35].

The direct antioxidant activity of NAC is due to the ability of its free thiol group to react with reactive oxygen and nitrogen species (RONS) [36]. Under experimental conditions, NAC reacts quickly with the hydroxyl radical (^•^OH), nitrogen dioxide (^•^NO_2_), carbon trioxide ion (CO_3_^•^^−^), and thiyl radical (RS^•^), in addition to the nitroxyl (HNO) that is the reduced and protonated form of nitric oxide (^•^NO) (Table 1). Reactions with the radical anion superoxide (O_2_^•^^−^), hydrogen peroxide (H_2_O_2_), and peroxynitrite (ONOO^−^) are relatively slow, as shown in Table 1 [34].

Under physiological conditions, the reaction rate of NAC is usually lower when compared to other enzymatic and nonenzymatic antioxidants, as well as other substrates, which raises doubts about the relevance of its direct antioxidant capacity against RONS, especially when other factors are considered as its endogenous concentration and location (intra- or extracellular) [3,11]. However, depending on its relative concentration compared to other thiols, it is possible that NAC has some direct antioxidant effect against some oxidative species, including ^•^NO_2_ and hypohalous acids (HOX) [3].

Like other thiols, experimental studies show that NAC through its free thiol can also bind to active redox metal ions, such as the ones from transition metals, copper (Cu^2+^) and iron (Fe^3+^), and heavy metals, cadmium (Cd^2+^), mercury (Hg^2+^), and lead (Pb^2+^), forming complexes that are easily excreted by the body [34]. Although NAC is able to reduce levels of metal ions in cases of toxicity [45], clinical studies that evaluate its chelating properties are limited. It is still unclear whether NAC potentially acts as a chelator or whether the benefits found are predominantly related to its action as an indirect antioxidant via an increase in the intracellular tripeptide containing cysteine (l-γ-glutamyl–l-cysteinyl–glycine), better known as glutathione (GSH) [11].

GSH is the most abundant nonprotein thiol in the body and one of the main antioxidants responsible for maintaining cellular redox status, which, in addition to reacting directly with reactive species, acts as a cofactor or substrate for various antioxidant enzymes [3]. The intracellular concentration of cysteine is lower and, therefore, it acts as a limiting factor in the rate of GSH biosynthesis. This characteristic explains the role of NAC as a prodrug of cysteine and intracellular GSH [10].

The importance of NAC as a potent antioxidant is directly linked to its ability to increase levels of intracellular cysteine with subsequent increase in GSH (Figure 2). The isolated uses of both GSH and cysteine were not effective in raising GSH levels within cells, making NAC one of the major strategies to reduce the damage caused by oxidative stress in cases of xenobiotic intoxication, such as paracetamol or in pathologies related to GSH deficiency, through the maintenance of their levels in different tissues [1,34,46].

In addition, NAC is also able to break down thiol proteins (such as cysteinylated extracellular proteins), releasing free thiols with higher antioxidant capacity, which potentiate GSH biosynthesis. Another mechanism linked to the indirect antioxidant activity exerted by NAC is related to its reducing capacity. NAC is capable of restoring systemic pools of low-molecular-weight (LMW) thiols and reduced protein sulfhydryl groups, which are involved in the regulation of the redox state, as is the case with mercaptoalbumin, which is the main antioxidant present in plasma and extracellular fluids [3].

Due to its reducing capacity, NAC also carries out one of its crucial activities as a potent expectorant. NAC is known as a mucolytic agent because it is able to reduce the disulfide bonds in crosslinked mucous proteins, interrupting their binding to the ligand and modifying their structures, thereby reducing the viscosity and elasticity of the mucus [34].

Experimental studies also describe the ability of NAC to alter the structure and/or function of proteins with (1) a reduction in the binding of the angiotensin II receptor in vascular smooth muscle cells [47]; (2) alteration of tumor necrosis factor (TNF-α) receptor affinity for cytokines [48], and (3) a decrease in the capacity of transforming growth factor beta 1 (TGF-β1) binding to the type III transforming growth factor receptor (TβRIII) betaglycan [49].

Lastly, NAC also exerts anti-inflammatory activity by inhibiting the nuclear factor kappa-light-chain-enhancer of activated B cells (NF-κB), which plays a critical role in the inflammatory cascade and immune response involved in the response to oxidative stress. NAC blocks the translocation and nuclear activation of the transcription factor NF-κB, responsible for the regulation of proinflammatory gene expression [9,36]. NAC has been shown to suppress the release of inflammatory cytokines TNFα, interleukin (IL)-1β, and IL-6 in lipopolysaccharide-activated macrophages [50].

## 4. Clinical Indications

Figure 3 displays the therapeutic uses of NAC and Table 2 lists the performed clinical studies, highlighting doses, treatment time, and administration routes.

### 4.1. Lung Diseases

The pathogenesis and progression of respiratory diseases can be explained by inflammation and increased oxidative stress, with a consequent reduction in endogenous antioxidants, such as GSH [115,116]. NAC represents a promising therapeutic target for the treatment of these diseases, including chronic obstructive pulmonary disease (COPD), cystic fibrosis (CF), and idiopathic pulmonary fibrosis (IPF), due to its antioxidant, anti-inflammatory, and mucolytic properties already described, related to its ability to replenish the GSH intracellular pool and reduce mucus production and viscosity [55,116] (Figure 3).

Typically, NAC is administered orally, using 600–1200 mg tablets up to three times a day [11]. For the treatment of chronic diseases such as COPD, which requires long-term use, the maximum licensed dose is 600 mg/day, but doses above 600 mg/day are constantly used in different clinical trials. The safety profile of NAC is usually similar, both for low doses (≤600 mg/day) and for high doses (>600 mg/day). Studies with a dosage up to 3000 mg/day in respiratory diseases have shown that NAC is safe and well tolerated [27] (Table 2).

The clinical efficacy of NAC for the treatment of patients with COPD has been documented in several clinical trials (Table 2); however, the results are still controversial. COPD is characterized by chronic symptoms related to airflow obstruction determined by the inhalation of toxic agents, being frequent in smoking patients. The BRONCUS (bronchitis randomized on NAC cost–utility study) study, a phase III, double-blind, randomized, placebo-controlled trial, followed 523 COPD patients for 3 years. Patients were treated with 600 mg per day of oral NAC versus placebo. The study demonstrated that treatment with low dosage of NAC did not affect the rate of decline in forced expiratory volume in 1 s (FEV 1) or vital capacity, nor did it affect the rate of exacerbation or the health status of patients with COPD [51]. Another clinical trial with the same oral dosage of 600 mg of NAC per day also showed no impact on the evolution of respiratory quality of life or on the rate of exacerbations in patients with chronic bronchitis or mild to moderately severe COPD [52].

Unlike the results found in previous studies, the effect of high dose *N*-acetylcysteine on air trapping and airway resistance of chronic obstructive pulmonary disease, a double-blinded, randomized, placebo-controlled trial (HIACE) conducted with 120 patients with stable COPD (58 treated with 600 mg of oral NAC twice a day and 62 with placebo), demonstrated a significant improvement in the function of the small airways, in addition to a reduction in the frequency of exacerbation of the disease [53]. Similarly, the placebo-controlled study on efficacy and safety of *N*-acetylcysteine high dose in exacerbations of chronic obstructive pulmonary disease held (PANTHEON) with 1006 patients with moderate and severe COPD, treated with 1200 mg of oral NAC per day, resulted in a significant reduction in acute exacerbations of COPD in the treated group compared to the placebo group, especially in patients with moderate disease [54].

The same finding was not found in the randomized, double-blind, placebo-controlled trial that compared the use of high-dose NAC (1800 mg twice a day, totaling 3600 mg/day) versus placebo for 8 weeks in COPD patients. The trial, while showing that high doses are well tolerated when used for a short time, did not produce positive effects on general respiratory health, lung function, or circulating measures of systemic oxidative stress and inflammation [55].

According to Tse et al. (2013) [53], treatment with low doses of NAC and the use of FEV 1 as a parameter for the evaluation of COPD in some studies may be responsible for the lack of effectiveness of the antioxidant and anti-inflammatory treatment of NAC in these patients. In general, higher doses generate greater bioavailability of NAC, increasing its effects. Thus, the authors suggested that low concentrations of NAC (600 mg/day) are unlikely to produce significant antioxidant and anti-inflammatory effects in improving lung function, exacerbation rate or risk of hospital readmission. However, higher doses can reduce the effects caused by oxidative stress and inflammation in COPD, demonstrating a positive impact on the rate of exacerbation [117].

In this perspective, in order to better elucidate the results of clinical trials, different meta-analyses examined the effects of NAC on COPD, using, as criteria for evaluation, the total number of exacerbations, number of patients with at least one exacerbation, and decline in FEV 1 [56,57]. Shen et al. (2013) [56] showed that long-term treatment with high doses of NAC (>600 mg/day) can reduce exacerbations in COPD, with no effect on the FEV 1 rate. Another meta-analysis also showed that prolonged treatment with higher doses of NAC (≥1200 mg/day) prevents exacerbations of COPD, while lower doses (600 mg/day) may be effective only for patients with chronic bronchitis, without COPD and without airway obstruction [57].

A more recent systematic review [58] that included a sample of 2691 participants, randomly assigned to the NAC (*n* = 1339) and control (*n* = 1352) groups, showed that treatment with low-dose NAC (≤600 mg/day) or high-dose NAC (>600 mg/day), for a minimum period of 6 months, can safely reduce the risk of COPD exacerbation, without adverse effects. However, NAC therapy had no effect on the rate of exacerbation or the parameters of pulmonary function (FEV 1). Cazzola et al. (2017) [118] also demonstrated that the use of high doses of NAC (1200 mg/day) as adjunctive therapy in patients with COPD can reduce the chances of exacerbations of the disease. On the basis of the results available up to date, the Global Initiative for Chronic Obstructive Lung Disease [119] recognizes that mucolytic and antioxidant drugs, such as NAC, can be used as an adjunct therapy to reduce the risk of acute exacerbation of COPD, but in a limited way due to the lack of robust randomized controlled clinical trials (level of evidence B).

In relation to the treatment of CF, NAC is often used in view of its mucolytic properties, through the hydrolysis of the disulfide bonds responsible for the connection of the mucin molecules [120]. Consequently, since the 1960s, the drug has been approved as an inhalation therapy for patients diagnosed with CF, but its effects on improving lung function or reducing pulmonary exacerbations have not yet been fully established [121,122].

Thus, the findings regarding the inhalation of NAC in patients diagnosed with CF are especially focused on its effectiveness in eliminating mucus secretion. In a systematic review with meta-analysis that aimed to synthesize the existing knowledge between antioxidants and the pulmonary function of individuals with CF, the authors recommended the administration of NAC (primarily by inhalation) as an adjunct to treatment, as it became evident that oral supplementation of NAC exhibited beneficial effects in preventing deterioration of lung function, but not significantly. Conversely, when used by inhalation, NAC resulted in a significantly favorable effect on lung function in adults after 3 months, being kept after 9 and 12 months [59].

CF patients characteristically exhibit recurrent infections/inflammations of the respiratory tract, which is consistent with an increase in oxidative stress. Therefore, 18 CF patients were scheduled for treatment with high doses of NAC (600, 800, and 1000 mg/day) three times a day, for a period of 4 weeks. In the end, the authors concluded that the high doses were able to raise the levels of GSH in blood neutrophils and the neutrophil count in the airways, with a significant reduction in elastase activity, the results being justified by the role of NAC in inhibiting the NF-κB pathway, causing a reduction in IL-8. Doses used were well tolerated by patients, while only mild adverse effects were reported (heartburn, nausea, flatulence) [60].

Subsequently, the use of NAC was reported in a randomized clinical study, in which the effect of 4 weeks of oral supplementation of NAC (2400 mg/day) on the oxidative stress markers in adults with CF and chronic pulmonary infection was evaluated, where it was concluded that there was an improvement in pulmonary function, but without reductions in lipid peroxidation, malondialdehyde (MDA), and 8-isoprostane markers. Nevertheless, the treatment was seen as well tolerated, despite one of the 11 participants dropping out of the study due to stomach pain [61].

On the basis of its antioxidant and anti-inflammatory activities, NAC has been used in different clinical trials as an option to treat patients with IPF in combination with standard therapies [28]. IPF is a chronic, progressive interstitial lung disease, of unknown etiology, which has a high mortality rate [123]. It is characterized by worsening dyspnea and lung function, as a result of abnormal wound healing in the interstitial and alveolar spaces of the lung, associated with the intense proliferation of myofibroblasts, which results in fibrosis. Inflammation and oxidative stress are part of its pathogenesis and are associated with disease progression [124,125].

It is speculated that NAC may be an effective therapeutic resource by inhibiting oxidation and restoring redox balance in patients with IPF [28]. Nevertheless, in 1989, the study by Cantin et al. [126] showed that patients with IPF had a fourfold lower concentration of GSH when compared to healthy individuals, suggesting that GSH deficiency in these patients contributes to the oxidative–antioxidant imbalance of the lower respiratory tract, causing greater damage to the cells of the lung parenchyma. In the same study, the authors suggested that restoring GSH levels in patients with IPF could be a rational therapeutic approach.

A clinical trial evaluated the effectiveness of inhaled NAC monotherapy in Japanese patients with mild to moderate IPF for a period of 48 weeks. Patients were randomly assigned to the NAC (*n* = 51) and control (*n* = 49) groups. The treated group received inhalation twice a day, with 352.4 mg of NAC diluted with saline, to a total volume of 4 mL, through MicroAir nebulizers and vibrating mesh technology. The control group did not receive any treatment or placebo during the research. Of the 100 participants, 76 completed the study (38 in each group). There were no significant general differences in the change in forced vital capacity (FVC) between the groups. Changes in other parameters, such as changes in lower arterial oxygen saturation, 6 min walking distance test (6MWD), and abnormal pulmonary function parameters (PFT) also did not show significant differences between the NAC and control groups. Regarding the safety of inhaled NAC, there were no significant differences in the number of adverse events reported for both groups, demonstrating that, in general, NAC was well tolerated. Bacterial pneumonia, cough, sore throat, and hypercholesterolemia were the most common adverse effects reported by participants [25].

Rogliani et al. (2016) [62], in their systematic review and meta-analysis, analyzed the efficacy and safety of drugs frequently used in IPF, pirfenidone and nintedanib, in addition to NAC, in 3847 patients (2254 treated and 1593 placebo). The study showed that both pirfenidone and nintedanib, but not NAC, were significantly effective in reducing the progression of IPF. In addition, the study also d attention to the safety of NAC (concentrations of 704.8 mg/day inhaled at 1800 mg/day in tablet form), suggesting a higher risk of adverse events, despite insignificant results.

Another recent meta-analysis [28], which included 21 studies published between 2005 and 2016, assessed the efficacy and safety of NAC therapy in IPF. Of the 1354 patients, 695 received NAC alone orally or inhaled or combined with other medications (commonly corticosteroids and pirfenidone) and 659 received other therapies. The commonly used oral dose was 1800 mg per day and the inhalation dose was 704.8 mg. Analysis of the data showed that NAC can decrease the decline in lung function in patients with IPF, related to the reduction in the decline in forced vital capacity (FVC) and in the diffusion capacity of carbon monoxide (DLCO), with slow disease progression due to stabilizing arterial oxygen partial pressure (PaO_2_). Additionally, the administration of NAC also improved the results of the 6 min walking distance test, corroborating the results of Sun et al. (2016) [123].

### 4.2. Cardiovascular Diseases

As with other diseases, oxidative stress is involved in the development and progression of cardiovascular disease (CVD). Clinical and especially experimental studies have shown the effectiveness of NAC in improving cardiac function, highlighting its cardioprotective role in conditions such as diabetic cardiomyopathy (DCM), acute myocardial infarction (AMI), heart failure (HF), and coronary artery disease (CAD) [127].

The multicenter clinical trial NACIAM (*N*-acetylcysteine in acute myocardial infarction), conducted with patients with ST-segment elevated myocardial infarction (STEMI) undergoing primary percutaneous coronary intervention, demonstrated that the combined treatment of high doses of intravenous NAC (20 mg/min in the first hour and 10 mg/min in the remaining 47 h) combined with a low dose of nitroglycerin (2.5 Pg/min for 48 h) was effective in reducing the size of acute infarction in these patients. The study also demonstrated that NAC improves myocardial rescue and acts in the faster resolution of chest pain in these patients, presenting itself as an adjunctive therapeutic proposal with a potential beneficial effect on primary percutaneous coronary intervention [128].

Other studies have also demonstrated positive effects on this therapeutic association [63,129], suggesting that NAC, in addition to acting as a potent antioxidant, potentiates the effects of nitroglycerin. In the study by Arstall et al. (1995) [63], the combined treatment of 15,000 mg of intravenous NAC over 24 h with nitroglycerin and streptokinase reduced oxidative stress in patients with AMI, measured by the higher proportion of reduced and oxidized glutathione (GSH/GSSG) in patients treated with NAC, associated with lower plasma malondialdehyde (MDA) levels and better preservation of left-ventricular function. In the study, all patients received an infusion of 5 μg/min of nitroglycerin before the start of the clinical trial. The treated group received an intravenous infusion of NAC, at 20 mg/min in the first hour, followed by 10 mg/min in the subsequent 23 h, totaling a dose of 15,000 mg in 24 h. After the initiation of NAC infusion, 100 mg of hydrocortisone was also administered, followed by 1.5 MU of intravenous streptokinase for 30 min. The control group did not receive the NAC infusion. Similarly, another study showed that the combined administration of intravenous NAC (15,000 mg/24 h), with nitroglycerin and streptokinase, significantly reduced plasma MDA levels and improved left-ventricular function in patients with AMI [64].

The use of NAC has also been explored in patients undergoing cardiac surgery. Studies have suggested that NAC is able to reduce postoperative complications in these patients, by reducing oxidative stress and inflammation [130,131]. However, a recent systematic review with 29 clinical trials and 2486 participants showed that NAC in different routes of administration (intravenous, oral, oral plus intravenous) and doses (<100, 100 to <300, ≥300 mg/kg/day) did not demonstrate significant efficacy in reducing the main adverse outcomes associated with cardiac surgery, such as mortality, acute renal failure, heart failure, length of stay and/or intensive care unit, arrhythmia, and acute myocardial infarction [65].

Another meta-analysis [66] also found no evidence that intravenous and/or oral administration of NAC in different therapeutic regimens (ranging from 50 mg/kg to 600 mg doses) in 578 patients had an effect on mortality. However, treatment with NAC reduced postoperative atrial fibrillation, which is a common type of arrhythmia after cardiac surgery and which can have an impact on the length of hospital stay of patients. Ozaydin et al. (2008) [67] also demonstrated that the rate of atrial fibrillation in the group treated with NAC, administered intravenously in doses of 50–150 mg/kg, 1 h preoperatively and 48 h postoperatively, was lower when compared to the placebo group. It is speculated that atrial fibrillation is related to increased oxidative stress and chronic inflammation due to GSH depletion [132]. Thus, current data reveal that NAC in patients after cardiothoracic surgery is effective in preventing arrhythmias, requiring further randomized clinical trials to assess other outcomes related to postoperative complications [133].

### 4.3. Psychiatric Illnesses

NAC has been recognized as a potential therapeutic strategy for psychiatric illnesses, including compulsive disorders, schizophrenia, bipolar disorder (BD), and depression. Its impact on the reduction of inflammatory cytokines, associated or not with oxidative processes, helps to explain the mechanism via which NAC can modulate the symptoms of psychiatric disorders [134]. A review based on current pieces of evidence [135] suggests that the etiology and progression of psychiatric disorders involve the deterioration of cerebral energy metabolism, mitochondrial functions, and redox imbalance, associated with environmental and genetic factors that have a clinical role not only in the characteristic symptoms of the disease, but also in the alteration of the circadian and metabolic rhythms of the individuals.

Schizophrenia is a psychiatric disorder characterized by positive symptoms, such as delusions and hallucinations, and negative symptoms, such as withdrawal from the environment, avolition, and blunted affection [135]. Clinical trials conducted with schizophrenic patients have shown that NAC is a safe and effective adjuvant therapy to improve the clinical outcomes of chronic schizophrenia [68,69,136].

A multicenter trial [68] evaluated the safety and efficacy of oral supplementation of 2000 mg of NAC per day associated with treatment with antipsychotic medication and demonstrated moderate benefits for treatment with NAC, which reduced the clinical severity measured by the Clinical Global Impression (CGI) scale and the Positive and Negative Syndrome Scale (PANSS). Additionally, Sepehrmanesh et al. (2018) [69] showed that the administration of 1200 mg of NAC had a positive impact on the positive, negative, general, and total psychopathological symptoms analyzed by PANSS, along with an improvement of cognitive performance. A meta-analysis also supported that NAC administered for a period of 24 weeks or more improves the symptoms of schizophrenia, as well as the cognitive domain of working memory, where studies had a daily dose range of 600 mg to 3600 mg [70].

Other clinical studies have analyzed the effect of NAC supplementation, based on glutamatergic neurotransmission via glycine reuptake at the *N*-methyl-d-aspartate (NMDA) receptor as a possible therapeutic target for the improvement of schizophrenia symptoms [137,138,139,140]. It is known that glutamate is an excitatory neurotransmitter in the brain, whose dysfunction in physiological control can contribute to the development of schizophrenia and other neuropsychiatric disorders. Evidence shows that the symptoms, cognitive deficits, and neurophysiological indices of schizophrenia can be reproduced by blocking these receptors [141]. NMDA is a receptor sensitive to redox reactions, inhibited by increased oxidative stress. Thus, the role of NAC as a precursor to GSH is to regulate the redox system and favor the function of the NMDA receptor [142].

In fact, the increases in oxidative stress in blood, cerebrospinal fluid in postmortem samples, and particular deficits in the glutathione system (unregulated GSH redox, GSH peroxidase and GSH reductase enzymes) are closely linked to the pathophysiology of schizophrenia and, by association, with etiopathology, as well as with worsening of symptoms to functional consequences [143]. A randomized clinical trial aiming to evaluate the impact of supplementation of 2700 mg/day of NAC for 6 months in patients with schizophrenia in standard treatment (antipsychotics, mood stabilizers, and/or benzodiazepine) identified that the supplementation was not able to improve the symptomatic and functional results, but increased the levels of glutathione in the brain and neurocognition (i.e., processing speed), while there was also a blood increase in GSH peroxidase activity [71].

Among mood disorders, BD and major depressive disorders (MDD) are the most prevalent. For BD, the literature suggests that mitochondrial dysfunction is involved in its pathogenesis. Thus, interventions that have an effect on mitochondrial function can act in the clinical improvement of depressive symptoms in BD. However, a randomized, placebo-controlled clinical trial conducted with 181 participants failed to demonstrate these effects from supplementing 2000 mg/day of NAC alone or combined with other antioxidants in patients with BD and depressive symptoms [72].

In MDD, inflammatory and oxidative processes seem to be part of the pathophysiology of depression. Inflammation markers, proinflammatory cytokines, acute phase proteins, and adhesion molecules are usually elevated in patients with this disorder, contributing to neuroinflammation that appears to be associated with structural and functional abnormalities in the brain and a lower therapeutic response [144]. Although NAC is a strong candidate for adjunctive treatment of depression, oral supplementation of 1000 mg/day (2 capsules of 500 mg twice daily) associated with the usual treatment of the disease, when compared to placebo, did not produce positive effects on treatment of individuals with MDD (moderate to severe) at the end of 12 weeks. However, secondary analysis revealed a more promising effect among patients with more severe depression [74]. Magalhães et al. (2011) [73] managed to demonstrate interesting results with the oral administration of 1000 mg/day in a small sample of patients (14 individuals) with BD II related to complete remission of depressive and manic symptoms in the treated group.

Additionally, Kishi et al. (2020) [75], when conducting a systematic review with meta-analysis from clinical, randomized, and double-blind studies evaluating NAC as an adjunct treatment in cases of bipolar depression and MDD, concluded that, despite NAC being able to reduce the score on the Global Clinical Impression Severity Scale, no significant improvement in symptoms was found.

Lastly, it is noted that the recommendations for the use of NAC as an adjunct therapeutic strategy in most psychiatric disorders are still limited. A clinical trial that aims to investigate the clinical efficacy of NAC supplementation in patients with MDD associated with its potential effects on inflammation and oxidative stress is ongoing (ID ClinicalTrials.gov NCT02972398).

### 4.4. Neurodegenerative Diseases

NAC has neurogenic and neuroprotective properties, since, unlike the administration of GSH and l-cysteine, NAC has the ability to effectively cross the blood–brain barrier, raising GSH levels in the brain, exhibiting anti-inflammatory actions, and acting on essential neurotransmitter systems, in addition to acting on glutamate [145]. Another aspect of this drug is the modulation of NMDA, which, when modified (hyperactivity or hypofunction), is responsible for neurological and psychiatric changes [146].

Taking into account oxidative damage, glutathione depletion, and/or NMDA receptor dysregulation in Parkinson’s disease, Alzheimer’s disease, and multiple sclerosis, high levels of RONS markers (lipid hydroperoxides, MDA, and superoxide dismutase activity—SOD) were found in Parkinson’s disease patients, along with reduced catalase activity. Furthermore, it has been suggested that MDA may be used as a biomarker in this disease, while SOD and lipid hydroperoxides may be associated with late disease characteristics [147].

In this sense, alternative therapies that elevate GSH and act on oxidative stress are often tested as modifiers of these diseases, such as NAC supplementation [148]. Thus, a study administering a single intravenous dose of NAC (150 mg/kg) in subjects with Parkinson’s disease revealed an increase in the blood GSH/GSSG and cerebral GSH ratio [76]. Similarly, 42 patients with Parkinson’s disease were tested with weekly intravenous doses of NAC (50 mg/kg) and higher oral doses (500 mg twice a day) for 3 months, in which the binding of the dopamine transporter was measured via brain images. The results suggested that supplementation could positively affect the patients’ dopaminergic system, which in turn had improved clinical effects [77]. However, intravenous administration is not sustainable in the long term; thus, a controlled clinical trial administered oral doses of 6000 mg/day in individuals with Parkinson’s, where they observed blood increases in antioxidant capacity (GSH/GSSG and catalase), but not in GSH cerebral level, possibly reflecting the low oral bioavailability when compared to the intravenous application [78].

In Alzheimer’s disease, NAC supplementation has been used as a plausible alternative for a future drug associated with a nutraceutical formulation, since, in several studies, the effect of NAC was evaluated in conjunction with other substances [149]. In a placebo-controlled clinical trial, individuals with mild cognitive impairment consumed 600 mg of NAC (associated with folate, alpha-tocopherol, vitamin B12, *S*-adenosylmethionine, and acetyl-l-carnitine) daily for 6 months and obtained improvement in the assessment scale of dementia and preservation of cognitive function [79].

NAC supplementation benefits those who suffer from multiple sclerosis (MS), a neurological disorder associated with white- and gray-matter lesions, with inflammation and oxidative damage mediated immunologically, specifically via reduced brain levels of GSH [150,151,152]. From this perspective, studies have suggested that antioxidant interventions can attenuate the frequent neurodegenerative processes in the disease [153]. Thus, a study with NAC and patients with MS identified improvements in cerebral glucose metabolism, cognition, and attention after administration of intravenous doses up to 50 mg/kg (once a week) and oral doses up to 1000 mg/day (6 times a week) [81]. In another study, randomized patients were supplemented with 1250 mg of NAC, in which no significant results were found in fatigue reduction outcomes, as well as changes in GSH levels in individuals with multiple sclerosis [80].

Therefore, more studies are needed to understand the effects of NAC in neurodegenerative diseases, as a function of the metabolic level, dosages, frequencies, route of administration, and clinical markers.

### 4.5. Liver Diseases

Acute and chronic liver diseases are highly prevalent worldwide, representing an important cause of morbidity and mortality [154]. Oxidative stress is a crucial factor in the pathogenesis of these diseases, participating in the liver’s fibrogenic response and stimulating its progression [155]. The efficacy of NAC in modulating inflammation and oxidative stress in liver diseases has already been analyzed through a wide systematic review of experimental and clinical studies that evaluated the antioxidant and anti-inflammatory roles of NAC in reducing liver damage [9].

NAC is widely accepted as a safe and effective drug to treat paracetamol poisoning. Its pathogenesis is already well understood and involves toxic effects on the liver due to the excessive production of *N*-acetyl-*p*-benzo-quinone imine (NAPQI), which is a hepatotoxic metabolite that under normal conditions is easily inactivated by liver GSH. However, in cases of intoxication, GSH is depleted and NAPQI accumulates in the organ, causing cell injury and death. The function of NAC is to replenish the stocks of GSH and contribute to a greater supply of oxygen to the injured liver [84,156].

More recently, experts produced an updated evidence-based guide to help clinical practice in the face of episodes of paracetamol poisoning, which is the most common cause of severe acute liver injury in Western countries (Box 2). According to the new guidelines, two bags of NAC administered intravenously (200 mg/kg in 4 h, then 100 mg/kg in 16 h) has a similar efficacy to the previous recommended dosage (of three bags), with the advantage of significantly reducing adverse reactions. The protocol also established a weight limit of 110 kg, with a maximum dosage of intravenous NAC equivalent to 22 g in the first infusion and 11 g in the second [82].

Box 2Current recommendation for the treatment of paracetamol poisoning.Management of paracetamol poisoningNew recommendation for standard regimen of two acetylcysteine bags ^ab^Initial infusion
▪NAC 200 mg/kg (maximum 22 g) in glucose 5% 500 mL (child, 7 mL/kg up to 500 mL) or sodium chloride 0.9% 500 mL (child, 7 mL/kg up to 500 mL) intravenously, over 4 h.
Second NAC infusion
▪NAC 100 mg/kg (maximum 11 g) in glucose 5% 1000 mL (child, 14 mL/kg up to 1000 mL) or sodium chloride 0.9% 1000 mL (child, 14 mL/kg up to 1000 mL) intravenously, over 16 h ^b^.
If ongoing NAC is required, continue at the rate of the second infusion (i.e., 100 mg/kg over 16 h). Higher ongoing infusion rates (i.e., 200 mg/kg over 16 h) may be required for massive paracetamol ingestions and a clinical toxicologist should be consulted ^c^.^a^ NAC is also compatible with 0.45% saline + 5% dextrose. ^b^ For adults (aged ≥14 years), dosing should be based on actual body weight rounded up to the nearest 10 kg, with a ceiling weight of 110 kg. For children (aged < 14 years), actual body weight should be used. ^c^ If the initial paracetamol concentration was more than double the nomogram line following an acute ingestion, acetylcysteine dose should be increased to 200 mg/kg (maximum 22 g) in glucose 5% 1000 mL (child, 14 mL/kg up to 1000 mL) or sodium chloride 0.9% 1000 mL (child, 14 mL/kg up to 1000 mL) intravenously, over 16 h. Adapted from Chiew et al. (2020) [82].

Like paracetamol poisoning, other drugs and herbal or dietary supplements can also cause acute or chronic liver damage. Liver damage induced by non-paracetamol drugs also includes direct, immune-mediated, and mitochondrial cell damage; however, the mechanism underlying hepatotoxicity does not involve GSH depletion and remains unclear [83,84]. Even so, it is suspected that NAC has beneficial effects in this situation by optimizing the oxygen supply and improving systemic hemodynamics [83].

A prospective study showed a significant improvement in survival of patients with acute liver failure treated with intravenous NAC (150 mg/kg over 1 h, followed by 12.5 mg/kg/h for 4 h and continuous infusion of 6.25 mg/kg/h for the remaining 67 h), with a lower mortality rate and shorter hospital stay [80]. Another study failed to demonstrate a beneficial effect associated with the use of intravenous infusion of NAC for 72 h in patients with non-paracetamol drug-induced liver injury [81]. 

The administration of intravenous NAC in patients with acute liver failure not induced by paracetamol shortly after admission reduces mortality and the need for liver transplantation, in addition to reduced encephalopathy, hospitalization, admission to the intensive care unit and failure of other organs [157]. Corroborating these data, a recent meta-analysis also concluded that NAC improves survival, transplant-free survival, post-transplant survival and length of hospital stay for patients with acute liver failure unrelated to paracetamol [158]. 

Nonalcoholic fatty liver disease (NAFLD) can manifest as simple steatosis or progress to chronic liver damage, due to the increased hepatic flow of free fatty acids, which generates an increase in oxidative stress concomitant with the suppression of intracellular antioxidant activity. Evidence from experimental studies has revealed that NAC blocks the accumulation of liver lipids and reduces proinflammatory markers, such as IL-6 and IL-1β, TNF-α, and NF-κB [159].

Clinical studies that analyze NAC supplementation in the liver function of patients with NAFLD are still limited. Khoshbaten et al. (2010) [85] demonstrated that oral administration of NAC (600 mg for 12 h) for 3 months resulted in decreased levels of alanine transaminase and spleen size in the group treated with NAC compared to patients receiving vitamin C. The study suggests that the decrease in fatty infiltration may be related to the reduction of the spleen and that the longer follow-up of these individuals could bring better results.

In alcoholic liver disease, treatment with oral prednisolone for 28 days combined with intravenous infusions of NAC for 5 days (day 1: at doses of 150, 50, and 100 mg/kg in 250, 500, and 1000 mL of solution of 5% glucose over a period of 30 min, 4 h and 16 h, respectively; days 2 to 5: 100 mg/kg per day in 1000 mL of 5% glucose solution) did not improve patient survival at 6 months, but an improvement in survival at 1 month was evidenced among patients with severe acute alcoholic hepatitis combination therapy, compared to those who received prednisolone alone [86]. Another study also failed to demonstrate the benefit of using high doses (300 mg/kg of NAC, diluted in 5% glucose adjusted to 500 mL/day) of intravenous NAC associated with enteral nutritional support in patients with severe acute alcoholic hepatitis [87]. In this sense, more clinical trials are needed for clarifying the potential effects of NAC on hepatic involvement.

### 4.6. Kidney Diseases

Acute kidney injury is a frequent manifestation in patients after cardiac surgery, in which it presents multifactorial pathophysiology that includes especially hemodynamic factors, nephrotoxic drugs, and inflammation. Therefore, NAC is used as an alternative perioperative therapy in an attempt to reduce the risk of acute kidney injury associated with surgery [160,161]. In meta-analysis, it was found that the intravenous administration of NAC (150 mg/kg to 1200 mg) in chronic renal patients undergoing cardiac surgery was able to significantly reduce the incidence of acute kidney injury and adverse cardiac events [88]. However, in another systematic review with meta-analysis, aiming to assess whether perioperative administration of NAC reduces the risk of acute kidney injury associated with cardiac surgery, similar rates were observed in the occurrence of the risk of acute kidney injury associated with cardiac surgery, changes in creatinine serum, and hospital mortality rate, in both groups. Thus, the authors concluded that perioperative administration of NAC is not recommended as an alternative to reduce the risk of acute kidney injury associated with cardiac surgery [162].

On the other hand, in patients with chronic kidney disease in stage 5, the administration of 5000 mg of NAC in 5% glucose in a final volume of 50 mL, intravenously during the hemodialysis session, was sufficient to significantly improve vascular reactivity pressure during reactive hyperemia [89]. In addition, when evaluated in patients with stage 3 chronic kidney disease, 600 mg oral NAC supplementation was not able to alter serum creatinine or cystatin C [90].

Lastly, there is no current consensus/guidelines established on the use or not of NAC in patients diagnosed with kidney disease, due to the quality of the studies, standardization of doses, and determination of the ideal method of administration.4.7. Gastrointestinal Diseases.

### 4.7. Gastrointestinal Diseases

Gastrointestinal diseases are frequent, and their treatment represents a great challenge today. Therefore, given its ability to act as an antioxidant, during the processes of stress, infection, and inflammation, NAC has been widely used in different gastrointestinal clinical conditions, including in cases of infection by *Helicobacter pylori*, colon cancer, and inflammatory bowel diseases (IBDs).

*Helicobacter pylori* infection is one of the most frequent infections worldwide, being closely related to environmental, behavioral, and genetic factors [163,164]. Upon colonization, the bacteria initiate a local inflammatory response by breaking the mucous barrier and adhering to the gastric epithelium, producing nitric oxide and reactive oxygen species (ROS) [165]. Once persistent, inflammation can lead to pathologies such as chronic gastritis, gastric cancer, and peptic ulcer [166,167,168].

Thus, in an attempt at adjuvant treatment to antibiotics, NAC is gradually being used, whereas studies indicate that this drug may have a mechanism to prevent the formation and/or destruction of biofilms, formed by the bacteria as a way of survival in the acidic environment and protection against the action of antibiotics, in addition to reducing oxidative stress and inflammatory activity of the gastric mucosa caused by the toxins of *Helicobacter pylori* [169,170].

Thus, studies have suggested that the addition of NAC to different antibiotic regimens may increase the rate of eradication of the bacterium, when compared to therapies without NAC supplementation [91,92]. Therefore, reinforcing this assumption, it was observed that NAC may have a favorable effect in decreasing bacterial colonization and in preventing gastritis in individuals with initial or mild infection by *Helicobacter pylori*, in which these effects were reported after administration of NAC associated with conventional antibiotic therapy, within 6 weeks [171].

However, as described in a systematic review, there is a lack of standards and details of the results of the few randomized clinical trials performed, which hinders the accuracy of information regarding the safety and efficacy of the applicability of NAC associated with the use of antibiotics in *Helicobacter pylori*-infected individuals [172]. Supporting the findings of this meta-analysis, a later study that aimed to assess the use of NAC associated with first-line triple therapy did not report an additive effect on the rate of eradication of the bacterium, when associated with the two therapeutic regimens [173].

In relation to colon cancer, evidence has pointed to the possible protective role of NAC against the oxidative damage common to cancer cells, where long-term (>7 days) drug management with NAC is able to protect the colon mucosa and reduce the occurrence of colon cancer, associated with colitis, by reducing the damage provoked by nitrotyrosine and 8-oxoG [93]. Likewise, the effect of administering NAC in patients with gastrointestinal cancer who underwent major abdominal surgery was assessed, whereby offering 1200 mg/day of NAC, via schematic parenteral nutrition 2 days before surgery until 5 days postoperatively, was able to reduce MDA and urinary nitrate, as well as improve the ratio of reduced to oxidized glutathione (GSH/GSSG) [94]. Likewise, in a randomized clinical trial, the chemopreventive action of NAC (800 mg/day) was identified as an agent in colon cancer, by reducing the hyperproliferation of the colonic epithelium [95].

In addition, IBDs with their inflammatory and oxidative status can benefit from the administration of NAC, in view of the replenishment of the GSH pool and increasing the GSH/GSSG ratio. Therefore, in research with patients diagnosed with Crohn’s disease, NAC was able to suppress the secretion of metalloproteinase-2 and metalloproteinase-3 (which, when elevated, are directly related to the severity of the disease and may contribute to the alteration of the epithelial barrier), in addition to exerting a direct effect on its activity when secreted, by increasing the GSH/GSSG ratio [174,175]. Likewise, treatment in patients with ulcerative colitis has shown significant improvement in clinical parameters (stool frequency and consistency, nocturnal stools, visible blood in stool, fecal incontinence, abdominal pain, abdominal tenderness, and need for antidiarrheals, evaluated by the Modified Truelove–Witts Severity Index) and a reduction in serum proinflammatory cytokines [96].

Furthermore, evidence suggests that NAC supplementation, when performed correctly, may be able to improve distal intestinal obstruction syndrome, especially in older individuals; however, more studies need to be performed in order to assess dose, route of administration, and safety [176,177].

When damaged, the intestinal barrier can be directly associated with diseases such as bacterial infections, autoimmune diseases, and systemic inflammation. Thus, NAC can play a role in restoring the intestinal barrier by deactivating protein kinases activated by mitogens and cellular Src (c-Src), which, in conditions of inflammation, are responsible for inducing the dissociation of the intestinal junctions and, therefore, compromising the intestinal barrier [178].

Therefore, although the results are promising, there are still controversies, where, in some cases, NAC does not have enough evidence to be used, thus requiring more robust and reliable studies.

### 4.8. Infectious Diseases

NAC has a range of identified therapeutic potential applications, notably the treatment of various infectious diseases, by attenuating mediators of oxidative stress and inflammation. In terms of improvement effects with NAC, evidence by Wang et al. (2008) [97] showed that its supplementation (8000 mg/day) was able to decrease total bilirubin and aminotransferases and increase prothrombin activity in patients diagnosed with severe chronic hepatitis B. Likewise, when the effect of the drug was evaluated in patients with acute viral hepatitis, an improvement in transplant-free survival and a reduction in mortality were observed [157,179].

From this perspective, in addition to its antioxidant activity, NAC was able to increase antimycobacterial activity in human macrophages infected with *Mycobacterium tuberculosis* [180]. Accordingly, patients diagnosed with pulmonary tuberculosis supplemented with NAC (600 mg/day) had significantly reduced sputum, as well as improved radiological response, serum glutathione peroxidase level, and immune response [98].

New studies are being carried out in an attempt to assess the effects of NAC in patients affected by SARS-CoV-2, whether preventively and/or as an adjuvant. Thus, it is believed that NAC would be able to block the exacerbated production of the angiotensin II-converting enzyme, limiting the penetration of the virus into cells, as well as participate in oxidative stress and inflammation (especially in inhibiting NF-kB activation), mechanisms that would possibly reduce the severity of lung disease in these patients [181,182,183]. Currently, a clinical trial is being carried out with patients supplemented with 6000 mg/day of NAC associated with therapies prescribed for COVID-19, which aims to quantify the number of patients accurately extubated and/or transferred from the intensive care unit (ICU) and the number of patients discharged due to clinical progress (ID NCT04374461) [99].

Nevertheless, in a phase III controlled randomized clinical trial with patients diagnosed with severe COVID-19, supplementation of intravenous NAC associated with standard treatment is being carried out, where the authors believe that, in addition to exerting a mucolytic effect, NAC will be able to help in patient recovery through its antioxidant effects (IRCT20200509047364N3, at Iranian Registry of clinical trials) [184].

Chorioamnionitis is a common complication of pregnancy, characterized by acute inflammation of the amniotic and chorionic membranes due to a bacterial infection. It is associated with significant brain injury in newborns, periventricular leukomalacia, and cerebral palsy, related to cytokine storm and oxidative stress, which are characteristic conditions of the disease [100,185]. Considering the potential of NAC as a fetal and neonatal neuroprotective, a prospective, double-blind study conducted with the mother–child binomial during pregnancy and postpartum (11 mothers and 12 newborns) evaluated the pharmacokinetics (PK) of intravenous NAC in pregnant women with chorioamnionitis and their children, in addition to placental transfer. The study showed that NAC clearance is faster in pregnant women than in nonpregnant women. Placental transfer of NAC was also rapid and the rate of fetal clearance was slower, indicating that NAC has a prenatal therapeutic potential for these newborns at high risk of brain inflammation [100].

The same research group in a prospective double-blind study with 22 pregnant women with a clinical diagnosis of chorioamnionitis and 24 newborns (including two pairs of twins) sought to assess the safety of NAC in the pre- and postnatal maternal and child group. The study treated women before delivery with NAC (100 mg/kg/dose) or saline, administered intravenously every 6 h until birth. The newborns in the treated group received NAC (preterm: 12.5 mg/kg/dose; term: 25 mg/kg/dose) 6 h after the mother’s last dose and every 12 h for five doses. The study demonstrated an absence of significant adverse effects, with beneficial results for mother and child. In treated newborns, NAC restored normal cerebrovascular coupling between major brain vessels, reduced the proinflammatory vascular endothelial growth factor (VEGF), and increased the anti-inflammatory cytokine IL-1 receptor antagonist (IL-1Ra). Treated women also demonstrated lower levels of cytokines associated with endothelial activation and leukocyte recruitment in inflammation [185].

Another placebo-controlled clinical trial was carried out with 67 pregnant women at risk of preterm delivery due to the diagnosis of infection and/or intrauterine inflammation (Triple I), which is an important cause of adverse outcomes in newborns, including premature delivery. The pregnant women followed the following protocol: infusion of a loading dose of NAC (150 mg/kg) administered in 1 h, followed by continuous infusion of NAC (50 mg/kg) for 4 h, and infusion of 100 mg/kg of NAC in the next 16 h or until delivery versus infusion of dextrose (5%) in water (D5W) in the placebo group. The study showed that maternal intrapartum NAC infusion is feasible, safe, and effective in reducing the mortality rate and severe neonatal morbidity, such as bronchopulmonary dysplasia, in addition to improving Apgar scores at 1 and 5 min and reducing the need for resuscitation with positive pressure ventilation [101].

In children under 15 years of age, NAC was used in a placebo-controlled clinical trial as a therapeutic proposal for acute pyelonephritis, which is a kidney infection, usually caused by bacteria that cause urinary tract infections. The children received NAC orally for 5 days, according to their weight, in the following scheme: children with a body weight equal to or greater than 30 kg, 900 mg/day; children weighing between 8.5 and 30 kg, 600 mg/day; children weighing less than 8.5 kg, 70 mg/kg/day. The study did not demonstrate any benefits associated with the use of NAC in these patients. The authors suggested that the short period of administration of NAC is not able to cause reductions in inflammatory biomarkers present in this condition [102].

### 4.9. Cancer Prevention and Treatment

With its important pharmacological and biological activities, NAC has been investigated as an agent in the prevention and treatment of some types of cancer. Therefore, NAC was widely researched as an anticancer agent, for reducing cancer hostility, increasing the apoptosis of cancer cells, and reducing catabolism, mitochondrial dysfunction, inflammatory mediators, and oxidative stress [46].

Oxidative stress causes a glycolytic and catabolic state in the stromal tumor cells, in which catabolites are released, such as lactate [186], which in turn support mitochondrial metabolism in carcinoma cells through metabolic heterogeneity. Thus, this heterogeneity increases the proliferation of cancer cells, reduces apoptosis, and stimulates larger tumors with metastases [187,188].

Won et al. (2020) [103] developed a prospective, controlled multicenter study to assess whether inhalation therapy with NAC would improve the quality of life of patients undergoing radiation therapy. The participants were patients with head and neck cancer identified in 10 institutions, where the experimental group inhaled 2400 mg per day nebulized liquid NAC, for a period of 8 weeks from the beginning of radiotherapy. This study found no adverse effects associated with drug inhalation, and the main finding was an improvement in the patients’ quality of life, as there was a greater reduction in the use of analgesics up to one month after therapy in the experimental group when compared to the control group (*p* = 0.014). Nevertheless, symptoms such as xerostomia were significantly improved in the experimental group (*p* = 0.019) [103]; similar data were also confirmed in a randomized clinical trial, in which patients who underwent mouth rinsing with 2500 mg/day of NAC before and after radiotherapy had an improvement in saliva thickening and xerostomia [104].

A randomized clinical trial with patients detected with head and neck cancer revealed that transtympanic injections with NAC were able to prevent cisplatin-induced ototoxicity when applied before exposure, admitting that this protective effect is a response to NAC binding with the molecules of cisplatin, capturing free radicals [189].

Oxidative stress induces the expression of the stromal monocarboxylate transporter 4 (MCT4), a marker of catabolism frequently elevated in breast cancer, suggesting the transport of cancer-associated stromal catabolites to highly proliferative cancer cells. Therefore, as NAC tends to prefer cells with altered glucose, it is understood that cells with a high concentration of MCT4 would be more susceptible to the effects of the drug. Thus, a pilot study exposed that NAC expressed safety and biological activity in breast cancer, where it reduced the proliferation of cells and the expression of MCT4 [190].

Nevertheless, the identification of gastric cancer in early stages is extremely important for effective treatment and reduction of mortality. Thus, it was attested that NAC in an adjuvant way to other drugs can assist in this identification, where its efficacy and cost-effectiveness were proven as premedication during examination of upper gastrointestinal endoscopy, since it acts directly by breaking the links between the molecules and assisting in the elimination of mucus, limiting one of the obstacles in this type of exam, which is the high presence of this content on the mucosal surface [191,192].

In summary, according to several studies, NAC supplementation can reduce the proliferation, migration, and invasion of cancer cells in different types of cancer [193,194]. Thus, NAC remains a promising strategy for future therapies in the treatment and prevention of cancers.

### 4.10. Other Conditions

In addition to the situations described above, NAC can be used in situations such as substance abuse disorders, autism spectrum disorder, dermatological diseases (type I lamellar ichthyosis, bullous morphea, systemic sclerosis, toxic epidermal necrolysis, atopic dermatitis, acne), wound healing, insulin resistance, pre-eclampsia, and sickle cell anemia, among others [195].

Commonly, studies have indicated that NAC acts as a promising agent for the treatment of substance use disorders, whereas the drug has the ability to restore the transmission of the prefrontal glutamate from the nucleus accumbens, reducing relapses. Furthermore, a randomized clinical trial examining the efficacy of NAC, in adjuvant treatment for smoking cessation, found that NAC (1800 mg) associated with first-line treatment was able to significantly reduce exhaled carbon monoxide (*p* < 0.01); the intervention group showed no withdrawal symptoms, depression, or anxiety and had a significant decrease in the levels of the soluble tumor necrosis factor 2 receptor (sTNF-R2) [105].

Additionally, due to its involvement in the extracellular modulation of glutamate, NAC becomes plausible in the treatment of autism spectrum disorder, since glutamatergic dysfunction and oxidative stress seem to be linked to the disease. Therefore, a systematic review with meta-analysis assessed that NAC supplementation allowed a significant reduction in hyperactivity and irritability, as well as increased social awareness in children with autism spectrum disorder. However, the authors recommended that more studies need to be carried out to have an effective NAC recommendation [196].

Regarding male fertility, NAC has positive effects on spermatogenesis and testicular function, which are well understood, which allows recommending this drug in the treatment of diseases induced by oxidative stress in the male reproductive system [197]. Randomized clinical trials with 50 infertile men with asthenoteratozoospermia showed that 600 mg/day of oral NAC for 3 months significantly improved sperm count, motility, and normal morphology [106].

NAC has also been proposed for the treatment of patients with sickle cell anemia due to the relationship of the disease with the increase in oxidative stress, concomitant with a reduction in antioxidant systems, which includes GSH. A pilot study showed that NAC may be a beneficial therapeutic strategy for sickle cell patients, as oral administration of 1200 or 2400 mg of NAC/day for 6 weeks was able to reduce the expression of phosphatidylserine in the erythrocyte membrane, which is a marker of oxidative stress [107]. Another phase II double-blind randomized clinical trial also showed that the administration of oral NAC up to 2400 mg/day is well tolerated by patients with sickle cell anemia or thalassemia, producing beneficial effects such as the inhibition of dense cells, restoration of levels of GSH, and reduction in the number of vaso-occlusive episodes [108]. On the other hand, Sins et al. (2016) [198] failed to demonstrate positive effects of NAC treatment in patients with sickle cell disease compared to placebo. However, the authors claim a low adherence rate among participants that may have compromised the results of the survey.

As for pre-eclampsia, only two studies are described in the literature evaluating the supplementation of NAC in humans, where one identified an effect of increasing weight at birth and Apgar score [109], while the other found stabilizing effects of the disease in women with the severe form of early-onset disease and/or HELLP syndrome (hemolysis, elevated liver enzymes, low platelets) [110].

For gestational diabetes, clinical trials are not yet available. However, an experimental study with female mice with pregestational diabetes showed that administration of NAC during pregnancy was able to increase GSH levels, reduce ROS levels in the fetal heart, and prevent the development of congenital heart defects in the offspring of diabetes. Thus, the study suggested that the use of NAC may be a potential strategy (alone or combined with insulin) to prevent congenital heart defects in newborns of diabetic mothers [199]. From this perspective, a clinical trial that aims to analyze the results of pregnancy and the metabolic profile of women with gestational diabetes mellitus supplemented with oral NAC (600 mg/three times daily/for 6 weeks) is registered in the Cochrane Central Register of Controlled Trials (ID CN-02170464).

Although the current data are considered preliminary and there are few randomized clinical trials on the involvement of NAC in these pathologies, this drug can be a viable option in the treatment or act in an adjuvant way [200].

A pilot study with 35 patients diagnosed with metabolic syndrome evaluated the efficacy of NAC, where tablets containing 600 mg were administered twice daily for a period of 6 weeks. As a result, the individuals in the intervention group had insulin resistance parameters estimated by the homeostatic assessment model (HOMA-IR), highly sensitive C-reactive proteins (hsCRP), systolic blood pressure, and significantly reduced triglycerides. Thus, NAC can exert several beneficial effects on the disease through antioxidant, anti-inflammatory, and vasodilatory routes [111].

Nevertheless, NAC is used in some ocular conditions, for example, in cases of dysfunction of the meibomian gland, a type of blepharitis (inflammatory disease associated with itching, redness, and flaking and crusting of the eyelids), in which it generates ocular dryness. Therefore, Akyol-Salman et al. (2010) [113], in a clinical trial, evaluated the use of 5% NAC in the form of eye drops in patients with this manifestation, where, after 1 month of therapy, the individuals had significant improvements in itching, eye burning, foreign body sensation, and blurred vision. In another clinical trial, comparing the topical use of NAC with the use of the combination of steroid–antibiotic betamethasone–sodium sulfacetamide therapy, it was observed that, in conjunction with the hygiene of the eyelids, the administration of NAC appears to be as effective as a topical steroid–antibiotic combination, in patients with meibomian gland dysfunction [114]. In addition, a randomized clinical trial identified a significant effect after oral administration of 100 mg/day of NAC in patients with chronic blepharitis, where the authors described that the finding was due to the effect of NAC in blocking lipid peroxidation in this disease [112]. When compared to the use of artificial tears, topical formulations of NAC for 5% in patients undergoing dry eye treatment administered four times a day for 2 weeks was able to cause improvements in the reported symptoms [201].

## 5. Conclusions

NAC acts as a precursor to intracellular cysteine, increasing the pool of GSH, which in turn is essential in several diseases linked to oxidative stress as a consequence of glutathione depletion. Thus, although NAC is used in a range of diseases, further studies are needed with a view to clarifying adequate dosages and treatment protocols, aiming at an efficient and wide performance of NAC in the treatment of different pathologies.

## Figures and Tables

**Figure 1 antioxidants-10-00967-f001:**
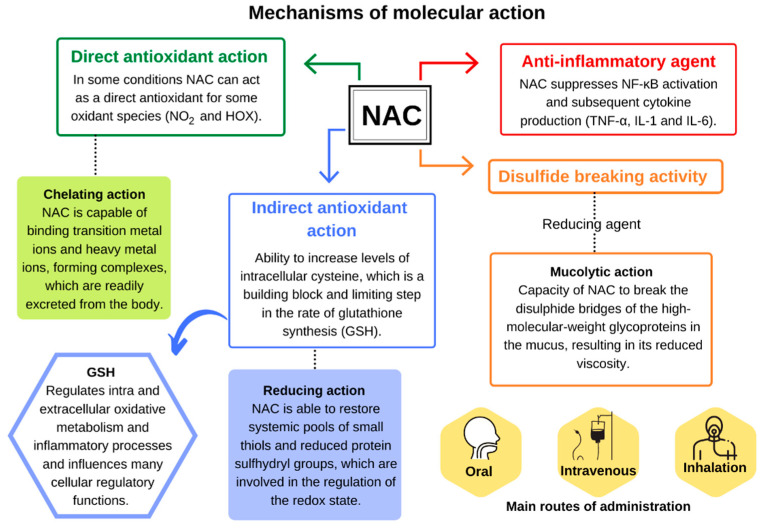
Mechanism of molecular action of *N*-acetylcysteine. Adapted from Aldini et al. (2018) [3].

**Figure 2 antioxidants-10-00967-f002:**
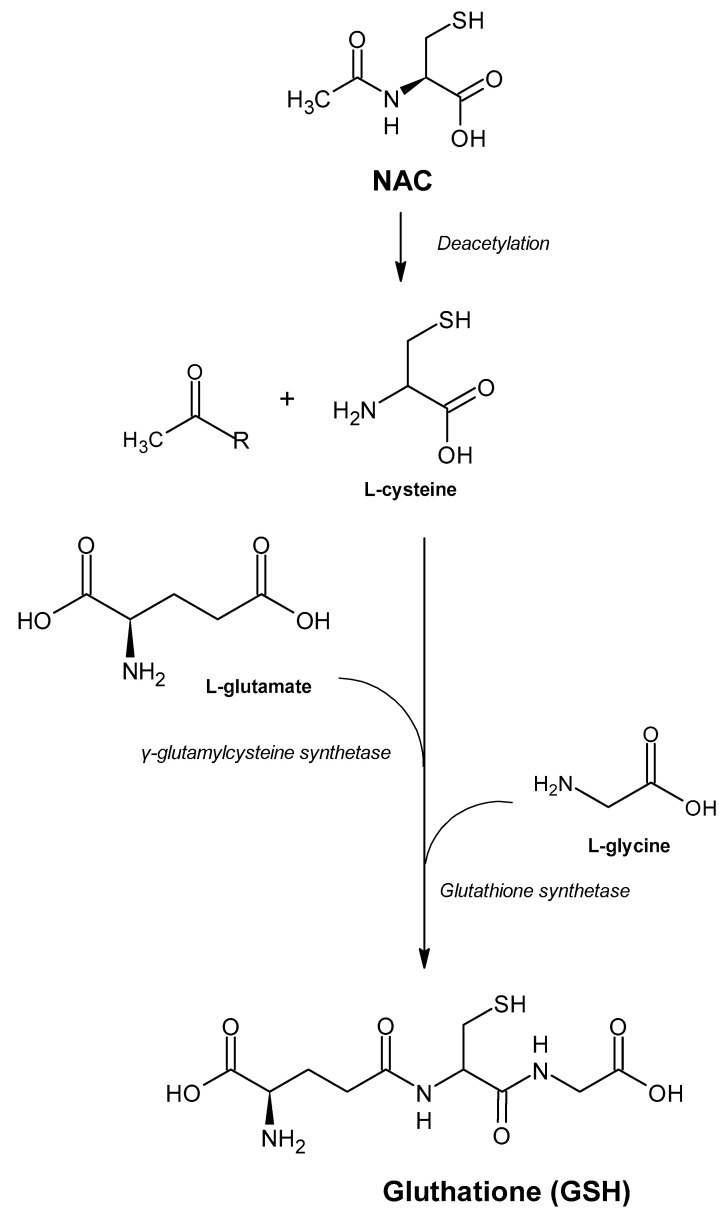
Chemical formula of *N*-acetylcysteine and its conversion to glutathione. Adapted from Rushworth and Megson (2014) [11].

**Figure 3 antioxidants-10-00967-f003:**
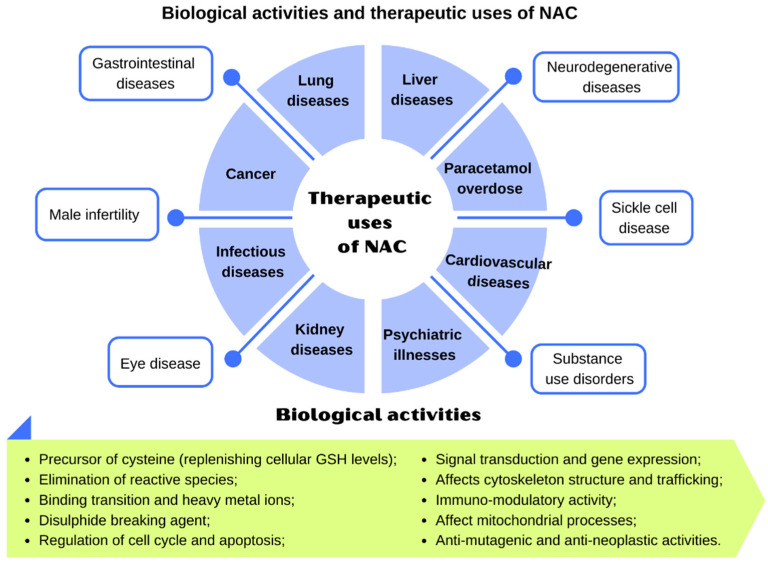
Biological activities and therapeutic uses of *N*-Acetylcysteine. Adapted from Samuni et al. (2013) [34].

**Table 1 antioxidants-10-00967-t001:** Rate constants of *N*-acetylcysteine reactions with various compounds under different experimental conditions.

Reactive Species	Rate Constant (M^−1^·s^−1^)	Experimental Conditions	Ref.
^•^OH	1.36 × 10^10^	pH 7, rt	[37]
^•^NO_2_	≈2.4 × 10^8a^ ≈1.0 × 10^7 b^	pH > pKa, rt pH 7.4, rt	[38] [39]
CO_3_^•^^−^	≈1.0 × 10^7^ 1.8 × 10^8^	pH 7, rt pH 12, rt	[40]
HNO	5 × 10^5^	pH 7.4, 37 °C	[41]
O_2_^•^^−^	68 ± 6 × 10^3^	pH 7, rt pH 7.4, 25 °C	[37] [42]
H_2_O_2_	0.16 ± 0.01 0.85 ± 0.09	pH 7.4, 37 °C pH 7.4, 25 °C	[37] [43]
ONOO^−^	415 ± 10	pH 7.4, 37 °C	[44]

**Table 2 antioxidants-10-00967-t002:** Studies carried out with *N*-acetylcysteine in different clinical conditions.

Disease	Study Type	Dose	Treatment Type	Administration Routes	References
Lung diseases					
Chronic obstructive pulmonary disease	Clinical trial	600 mg/day	3 years	Oral	[51]
	Clinical trial	600 mg/day	3 years	Oral	[52]
	Clinical trial NCT01136239	1200 mg/day	1 year	Oral	[53]
	Clinical trial ChiCTR-TRC-09000460	1200 mg/day	1 year	Oral	[54]
	Clinical trial NCT01599884	3600 mg/day	8 weeks	Oral	[55]
	Systematic review with meta-analysis	>600 mg/day	Long term	Oral	[56]
	Systematic review with meta-analysis	≥1200 mg/day	Long term	Oral	[57]
	Systematic review	Low doses: ≤600 mg/day High doses: >600 mg/day	Minimum of 6 months	Oral	[58]
Cystic fibrosis	Systematic review with meta-analysis	600 mg to 2800 mg/day	3.9 and 12 months	Oral or inhalation	[59]
	Clinical trial	1800, 2400 and 3000 mg/day	4 weeks	Oral	[60]
	Clinical trial 2007-001401-15	2400 mg/day	4 weeks	Oral	[61]
Idiopathic pulmonary fibrosis	Systematic review with meta-analysis	704.8 to 1800 mg/day	-	Oral or inhalation	[62]
	Systematic review with meta-analysis	Oral doses: 1800 mg/day Inhalation doses: 704.8 mg	3 to 15 months	Oral or inhalation	[28]
Cardiovascular diseases	Clinical trial	15,000 mg/day	24 h	Intravenous	[63]
	Clinical trial	15,000 mg/day	24 h	Intravenous	[64]
Cardiac surgery	Systematic review with meta-analysis	<100 mg/kg/ to ≥300 mg/kg/day	<24 and >48 h	Oral and/ or intravenous	[65]
	Systematic review with meta-analysis	50 mg/kg to 600 mg	Until 48 h	Oral and/ or intravenous	[66]
	Clinical trial	50–150 mg/kg	1 h preoperatively and 48 h postoperatively	intravenous	[67]
Psychiatric illnesses					
Schizophrenia	Multicenter clinical trial	2000 mg/day	4 weeks	Oral	[68]
	Clinical trial IRCT:2015080223463	1200 mg/day	12 weeks	Oral	[69]
	Systematic review with meta-analysis	600 mg to 3600 mg/day	≥24 weeks	Oral	[70]
	Clinical trial NCT01354132	2700 mg/day	6 months	Oral	[71]
Bipolar disorder	Clinical trial ACTRN12612000830897	2000 mg/day	16 weeks	Oral	[72]
	Clinical trial 12605000362695	1000 mg/day	24 weeks	Oral	[73]
Depression	Clinical trial ACTRN12607000134426	1000 mg/day	12 weeks	Oral	[74]
	Systematic review with meta-analysis	1000 mg to 3000 mg/day	-	Oral	[75]
Neurodegenerative diseases					
Parkinson’s disease	Clinical trial NCT01427517	150 mg/kg	1 h	Intravenous	[76]
	Clinical trial NCT02445651	Intravenous doses: 50 mg/kg Oral doses: 1000 mg/day	3 months	Oral and intravenous	[77]
	Clinical trial NCT02212678	6000 mg/day	4 weeks	Oral	[78]
Alzheimer’s disease	Clinical trial NCT00903695	600 mg/day	6 months	Oral	[79]
Multiple sclerosis	Clinical trial NCT02804594	1250 mg/day	4 weeks	Oral	[80]
	Clinical trial NCT03032601	Intravenous doses: 50 mg/kg (once a week) Oral doses: 1000 mg/day (6 times a week)	2 months	Oral and Intravenous	[81]
Liver diseases					
Paracetamol poisoning	Guide clinical practice	200 mg/kg in 4 h, then 100 mg/kg in 16 h	20 h	Intravenous	[82]
Acute liver failure	Randomized case control study	150 mg/kg over 1 h, followed by 12.5 mg/kg/h for 4 h and continuous infusion of 6.25 mg/kg/h for the remaining 67 h	72 h	Intravenous	[83]
	Systematic review	150 mg/ kg over 1 h, followed by 12.5 mg/kg/h for 4 h, and continuous infusions of 6.25 mg/kg/h for the remaining 67 h	72 h	Intravenous	[84]
Nonalcoholic fatty liver disease	Clinical trial	600 mg/day	3 months	Oral	[85]
Acute alcoholic hepatitis	Multicenter clinical trial NCT00863785	Day 1: at a dose of 150, 50, and 100 mg/kg a period of 30 min, 4 h, and 16 h, respectively; Days 2 through 5: 100 mg/kg	28 days	Intravenous	[86]
	Multicenter clinical trial	300 mg/kg	14 days	Intravenous	[87]
Kidney diseases					
Chronic renal patients undergoing cardiac	Systematic review with meta-analysis	150 mg/kg to 1200 mg	6 h to 7 days	Intravenous	[88]
Chronic kidney disease	Clinical trial	5000 mg	One hemodialysis session	Intravenous	[89]
	Clinical trial	600 mg/day	2 doses with 1 week interval	Oral	[90]
Gastrointestinal diseases					
*Helicobacter pylori* infection	Clinical trial	600 mg/day	14 days	Oral	[91]
	Clinical trial	800 mg/day	5 days	Oral	[92]
Colon cancer associated with colitis	Systematic review	8000 mg to 1200 mg/day	7 days to 12 weeks	Oral	[93]
Gastrointestinal cancer	Clinical trial	1200 mg/day	2 days before surgery until the fifth day postoperatively	Schematic parenteral	[94]
	Clinical trial	800 mg/day	12 weeks	Oral	[95]
Ulcerative colitis	Clinical trial	800 mg/day	4 weeks	Oral	[96]
Infectious diseases					
Chronic hepatitis B	Clinical trial	8000 mg/day	28 days	Intravenous	[97]
Pulmonary tuberculosis	Clinical trial	600 mg/day	2 months	Oral	[98]
SARS-CoV-2	Clinical trial NCT04374461	6000 mg/day	3 weeks	Intravenous	[99]
Chorioamnionitis	Clinical trial	100 mg/kg/dose and 12.5–25 mg/kg/dose their infants	Diagnosis until delivery and every 12 h for 5 doses to their infants	Intravenous	[100]
Infection and/or intrauterine inflammation	Clinical trial NCT00397735	150 mg/kg administered in 1 h, followed by continuous infusion of NAC (50 mg/kg) for 4 h and infusion of 100 mg/kg of NAC in the next 16 h or until delivery	-	Intravenous	[101]
Acute pyelonephritis	Clinical trial NCT02080182	Children with a body weight equal to or greater than 30 kg, 900 mg/day, children weighing between 8.5–30 kg, 600 mg/day, and those weighing less than 8.5 kg, 70 mg/kg/day	5 days	Oral	[102]
Cancer prevention and treatment					
Radiation therapy, head and neck cancer	Prospective, controlled multicenter study	2400 mg/day	8 weeks	Inhalation therapy	[103]
	Clinical trial NCT02123511	2500 mg/day	-	Rinsing	[104]
Other conditions					
Smoking cessation	Clinical trial NCT02420418	1800 mg/day	12 weeks	Oral	[105]
Male fertility	Clinical trial IRCT20170830035998N4	600 mg/day	3 months	Oral	[106]
Sickle cell anemia	Clinical trial NTR1013	1200 or 2400 mg/day	6 weeks	Oral	[107]
	Clinical trial	600 mg, 1200 mg or 2400 mg/day	7 months	Oral	[108]
Pre-eclampsia	Clinical trial	400 mg/day	6 weeks	Oral	[109]
	Clinical trial	600 mg/day	-	Oral	[110]
Metabolic syndrome	Pilot study	1200 mg/day	6 weeks	Oral	[111]
Ocular condtions					
Chronic blepharitis	Clinical trial	100 mg/day	1 to 4 months	Oral	[112]
Meibomian gland dysfunction	Clinical trial	5%	1 month	Eye drops	[113]
	Clinical trial	5%	1 month	Eye drops	[114]

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
