# Peer review of "N-Acetylcysteine (NAC): Impacts on Human Health"

_antioxidants, 2021, doi:10.3390/antiox10060967_

Round 1

Reviewer 1 Report

This review on NAC is well written and organized and explores each aspect of NAC in human health.

Starting from the  biochemistry of this compound, the authors gave exhaustive information on current use, toxicity data, adminitration routes, population involved in the therapy, dosages, pharmacokinetics and bioavailability. 

Further data on the mechanism of action in each disease wer provided as well as clinical trials results.

Figures and tables completed the correctness of the paper which is suitable for this hournal after the following minor suggestions:

line 59: forms?

lines 133 and 141 are the same

line 140: I prefer "are available"

line 238: change to "which requires"

line 355: microar?

line 530: the ref is uncorrectly written

page 21 and 23: please italicize the bacterial species names

line 656: change to "bacteria initiate"

line 809: I prefer "similar data were also"

line 877: apgar?

Author Response

We appreciate your comments and suggestions. We accepted all your suggestions. Thank you.

Point 1: line 59: forms?

Response 1: We replace forms for generates.

Point 2: lines 133 and 141 are the same

Response 2: We excluded the repeated phrase.

Point 3: line 140: I prefer "are available."

Response 3: We corrected as indicated.

Point 4: line 238: change to "which requires."

Response 4: We corrected as indicated.

Point 5: line 355: microar?

Response 5: Fixed to "MicroAir." This is the nebulizer model used in the study.

Point 6: line 530: the ref is incorrectly written

Response 6: We carried out the correction.

Point 7: page 21 and 23: please italicize the bacterial species names

Response 7: We corrected as indicated.

Point 8: line 656: change to "bacteria initiate"

Response 8: We corrected as indicated.

Point 9: line 809: I prefer "similar data were also."

Response 9: We corrected as indicated.

Point 10: line 877: apgar?

Response 10: Fixed for "Apgar score." The method is used for the immediate evaluation of the newborn, mainly in the first and fifth minutes of life.

Reviewer 2 Report

The manuscript entitled " N-acetylcysteine (NAC): Impacts on Human Health” by Tenório et al. is an update overview of the medicinal effects and applications of NAC to human health. The review is well organized and it has a complete design.

I have some comments:

- A careful stylistic and linguistic revision is required (e.g. line 920).

- The authors could report in the table 2 about study type also the identifier number of the clinical trials.

-Figure 1 is too written, I suggest to summarize using few words or some rappresentation.

Author Response

We thank the reviewer and the careful analysis of our paper. Below are the answers to all the questions asked, as well as the details of the adjustments.

Point 1: A careful stylistic and linguistic revision is required (e.g., line 920).

Response 1: We appreciate your comments and suggestions, and to make it easier to understand, we have improved the writing.

Point 2: The authors could report in table 2 about study type also the identifier number of the clinical trials.

Response 2: It is a significant comment. We added all found identifier numbers. See marked column in table 2

Point 3: Figure 1 is too written; I suggest summarizing using few words or some representation.

Response 3: We appreciate your suggestions; we reduced the number of words. See modified figure.

Reviewer 3 Report

The review by Micaely Cristina dos Santos Tenório et al, describes the impacts on human health of N-acetylcysteine (NAC).

The manuscript is clearly written, original and of interest in its field.

 I recommend that the paper be accepted with just one minor revision:

  1. the authors in the introduction refer to the in vitro and in vivo studies with the NAC, but do not elaborate on this point. It would be very interesting for the reader to have an overall picture that also collects these experimental data such as for example:

10.1038/sj.bjp.0703421

10.1002/brb3.208

10.1186/s12917-020-2234-9

10.2353/ajpath.2010.091253

10.2147/NDT.S241497

10.5935/1518-0557.20180079

Author Response

Thanks a lot for the comments that allowed the improvement of the present article.

 Response 1: We appreciate your comment. We have cited some in vitro and in vivo studies as suggested. See added references (6-8).

  1. Cuzzocrea, S.; Mazzon, E.; Costantino, G.; Serraino, I.; Dugo, L.; Calabrò, G.; Cucinotta, G.; De Sarro, A.; Caputi, A.P. Beneficial effects of n-acetylcysteine on ischaemic brain injury. British Journal of Pharmacology. 2000, 130, 1219-1226. Doi: 1038/sj.bjp.0703421.
  2. Crupi, R.; Gugliandolo, E.; Siracusa, R.; Impellizzeri, D.; Cordaro, M.; Di Paola, R.; Britti, D.; Cuzzocrea, S. N-acetyl-L-cysteine reduces Leishmania amazonensis-induced inflammation in BALB/c mice. BMC Vet Res. 2020, 16, 13. https://doi.org/10.1186/s12917-020-2234-9.
  3. Poncin, S.; Colin, IM.; Decallonne, B.; Clinckspooor, I.; Many, MC.; Denef, JF.; Gérard, AC. N-acetylcysteine and 15 deoxy-{delta}12,14-prostaglandin J2 exert a protective effect against autoimmune thyroid destruction in vivo but not against interleukin-1{alpha}/interferon {gamma}-induced inhibitory effects in thyrocytes in vitro. Am J Pathol. 2010, 177, 219-28. DOI: 10.2353/ajpath.2010.091253.

The present review of NAC concerned NAC’s impacts on human health, as written in the introduction: “Thus, the purpose of this review is to provide an overview of the medical effects and applications of NAC to human health based on current therapeutic evidence.”